# Dynamic manipulation of droplets using mechanically tunable microtextured chemical gradients

Ali J. Mazaltarim[1], John J. Bowen[1], Jay M. Taylor[1] & Stephen A. Morin [1,2,3 ✉]

Materials and strategies applicable to the dynamic transport of microdroplets are relevant to surface fluidics, self-cleaning materials, thermal management systems, and analytical devices. Techniques based on electrowetting, topographic micropatterns, and thermal/chemical gradients have advanced considerably, but dynamic microdroplet transport remains a challenge. This manuscript reports the fabrication of mechano-tunable, microtextured chemical gradients on elastomer films and their use in controlled microdroplet transport. Specifically, discreet mechanical deformations of these films enabled dynamic tuning of the microtextures and thus transport along surface-chemical gradients. The interplay between the driving force of the chemical gradient and the microtopography was characterized, facilitating accurate prediction of the conditions (droplet radius and roughness) which supported transport. In this work, the use of microtextured surface chemical gradients in mechano-adaptive materials with microdroplet manipulation functionality was highlighted.

[1] Department of Chemistry, University of Nebraska – Lincoln, Lincoln, NE, USA. [2] Nebraska Center for Materials and Nanoscience, University of Nebraska – Lincoln, Lincoln, NE, USA. [3] Nebraska Center for Integrated Biomolecular Communication, University of Nebraska – Lincoln, Lincoln, NE, USA. ✉email: smorin2@unl.edu

The transport of liquid microdroplets on surfaces—a capability useful to, for example, surface fluidics and smart coatings—using chemical wettability gradients was initially demonstrated in the seminal work of Chaudhury and Whitesides[1]. This work fostered deeper understandings of surface gradients and inspired the development of new strategies for the transport of microdroplets on surfaces, which can now be achieved via chemical[1–5], thermal[6,7], surface charge density[8], and surface topographical gradients[9–12]. The fabrication and operation of materials or devices which adopt these design strategies typically requires multistep procedures (e.g., photolithography), external power/control systems, and/or optimized environments. These factors together can limit the application space in which these materials/devices are useful. Accordingly, recent developments have leveraged fundamental understandings of surface microdroplet transport[13] in an effort to realize new strategies for the design and fabrication of devices capable of increasingly dynamic and sophisticated manipulations of droplets[4,5,8,13–17]. Despite these efforts, dynamic transport technologies require further evolution to overcome key limitations: switching between transport states using chemical/structural gradients can be slow and difficult to accomplish (e.g., photoactivated surfaces are tied to the kinetics of surface reactions and structural gradients have highly specific transport/geometry relationships)[4,5,9–12], continuous transport using electrostatic systems requires additional processing and environmental conditions can impact performance[8], and in many cases, dedicated input systems (e.g., electrical power supplies, light sources, or vibrational stages) are required to stimulate/control transport[2,4,5,13,16].

We set out to design mechano-adaptive materials with dynamic microdroplet handling capabilities that were easy to fabricate following simple and scalable procedures that did not require additional processing to control/operate once made. Our strategy, which combines the action of chemical gradients with mechanically tunable surface microstructures on stretchable silicone films, simultaneously removes the requirement for additional chemical processing and highly refined modifications to microstructural patterns, providing a strategy for programmable droplet transport based on simple mechanical input. This approach significantly elevates the practical applications of programmable droplet transport systems, and yields environmentally stable systems capable of rapid/instantaneous switching of droplet transport following mechanical stimulation from a potentially diverse range of sources (e.g., biomechanical input, electromechanical systems, or environmental stressors). We believe our versatile approach which is devoid of static/rigid components represents an important advancement in adaptive/programmable droplet transport capabilities that will have relevant applications in soft materials (i.e., wearables), adaptive systems (i.e., soft robotics), water-harvesting technologies, and energy generation.

Silicone surfaces can be chemically activated using plasma or ultraviolet ozone (UVO) oxidation procedures[18,19]. These oxidation methods produce reactive silanol (Si-OH) groups and generate a stiff surface oxide layer[18,19]. The surface properties (i.e., wettability) of these films can be further tailored by reacting organosilanes with the surface silanol moieties[20]. When elastomer mechanics and surface chemistry are coupled, surfaces with mechano-dynamic molecular layers, so-called "mechanically assembled monolayers", have been realized[21]. The influence of these surface modification procedures on the surface microtopography of silicone films has been extensively investigated[22], and the formation of sinusoidal surface microtopographies (wrinkles) caused by the mechanical mismatch between stiff surface layers and the soft underlying polymer has been thoroughly examined[22–27]. Mechanically reversible surface topographies have been used to modulate surface wettability[25,28,29]

and adhesion[30,31] and in the fabrication of materials for stretchable electronics[32,33].

In a related line of research, chemical and topographical gradients have been used to impart surface energy gradients for droplet transport[1–5,9–12]. In the present work, we combined the effects of a surface chemical gradient and a dynamically tunable microtextured surface to realize programmable transport functionality. To explain our approach, we will briefly describe the physicochemical factors that govern droplet motion on gradients. Central to this discussion is the chemical driving force ($F_d$) which accounts for the free energy gradient (wettability gradient) acting on the droplet as it moves along the surface[2,3]. The driving force is defined as:

$$F_d \cong \pi R^2 \gamma \left( \frac{d cos\theta}{dx} \right) \tag{1}$$

where $\gamma$ is the surface tension of the liquid, $R$ is the droplet radius, and $\frac{d cos\theta}{dx}$ (which we will refer to as $\Phi$) parametrizes the change in contact angle ($\theta$) along the gradient. A hysteresis force ($F_h$) arises due to the contact angle hysteresis of the droplet, where $F_h$ can be approximated as:

$$F_h = 2\gamma R(cos\theta_r - cos\theta_a) \tag{2}$$

where $\theta_r$ and $\theta_a$ are the receding and advancing contact angles, respectively[2,3]. For convenience, we will express contact angle hysteresis ($cos\theta_r - cos\theta_a$) as $b$ in the following expressions. Thus, the net force ($F$) acting on the droplet on a smooth chemical gradient (where the roughness factor, $r$, is equal to 1) prior to the onset of motion can be expressed as[2]:

$$F = \pi R^2 \gamma \Phi - 2\gamma R b \tag{3}$$

Consequently, the action of the gradient on the droplets ($F_d$) must overcome the hysteresis force ($F_h$) before transport will occur[2,3]. The wetting state of a droplet on a wrinkled surface (where $r > 1$) depends on the wrinkle characteristics (amplitude, wavelength, etc.)[25] and for droplets in the Wenzel wetting state (observed in this work), contact angle hysteresis scales with surface roughness[34,35]. Thus, the net force becomes:

$$F = \pi R^2 \gamma \Phi - 2\gamma r R b \tag{4}$$

where roughness increases the magnitude of the hysteresis term. During droplet transport, a viscous drag force ($F_v$) acts against the microdroplet. Using the lubrication approximation and assuming a circular droplet profile, $F_v$ can be approximated as[2,3,9,12,36]:

$$F_v = 3\eta\pi R r V \int_{x_{min}}^{x_{max}} \frac{dx}{H} \approx 3\eta\pi R r V \ln\left(\frac{x_{max}}{x_{min}}\right) \tag{5}$$

where $\eta$ is the viscosity of the liquid, $v$ is the droplet velocity, and $H$ is the height of droplet. The integration limits $x_{min}$ and $x_{max}$ are the characteristic lengths of a droplet ($x_{min}$ is the length of a molecule, and $x_{max}$ can be represented as the radius of a droplet). The steady-state velocity of a liquid microdroplet moving along a wettability gradient can then be obtained by equating $F_d$ to $F_v$ and approximated according to the Greenspan model as[2]:

$$v \cong \frac{\gamma R}{\eta}(\Phi) \tag{6}$$

Although the above treatment does not consider the influence of contact angle hysteresis on droplet velocity[2], experimental observations loosely follow this approximation, and droplet velocity scales linearly with droplet radius[2,3,7]. However during incipient motion—the condition immediately prior to the onset of droplet transport critical to this work—$F_v$ is negligible and can be ignored[9,36].

The general force balance expressions (Eq. (3) and Eq. (4)) have guided fundamental understandings of droplet transport on

gradients, but their predictive abilities are somewhat limited. For example, the droplet radii which meet the predicted condition for transport (i.e., when $F > 0$ following Eq. (3) and Eq. (4)) are often lower than those observed experimentally[3,37,38]. Other forces (e.g., friction[37–40], capillarity[36,41], and hydrodynamics[3]) have been cited as possible causes for this discrepancy. These explanations hinted at a delicate force balance that could be used for experimental control, and we hypothesized that dynamic control of surface roughness and thus the hysteresis force could enable active control of droplet transport along chemical gradients in ways not yet realized.

Previously, we demonstrated a method—one based on template-guided vapor diffusion—for the generation of periodic chemical gradients on soft surfaces[42]. Further, we have employed strain-induced, irreversible microtexture to synthesize wettability patterns and gradients[43], and we have applied mechanical buckling to generate reversible microtexture for optical functionality[44]. We combined our unique expertize in the generation of periodic gradients and the generation of reversible microtextures to the synthesis and operation of chemical gradients with mechano-dynamic microtexture.

In this work, we fabricate chemical gradients on the surface of pre-strained silicone films[42], which spontaneously form wrinkled microtextures when pre-strain is released[22–27]. Specifically, we use polydimethylsiloxane (PDMS)—a soft silicone elastomer with tunable surface chemistry and microtopography—as the elastomer support for adaptive microdroplet transport. PDMS can be stretched manually or with simple tools (e.g., electromechanical devices) allowing for several convenient methods to apply mechanical tension. We treated pre-strained ($\varepsilon_{pre} = 0.20$ or 20%) PDMS films with UVO—an oxidation method capable of generating micron thick surface silica layers—to induce the formation of controlled microwrinkles following strain release[25–27]. The generated wrinkles can be accurately modelled using well established models (see Supplementary Discussion, section 1)[10,24–27]. The films are briefly exposed to oxygen plasma following UVO treatment to increase surface silanol concentration for subsequent functionalization. We then utilize mechanical strain to control the pitch/amplitude of these wrinkled microtextures reversibly, thus enabling dynamic control of the forces which govern microdroplet transport along the chemical gradient. In addition, we adapt the force balance equations used to describe microdroplet motion on chemical gradients to predict the conditions (surface roughness and critical droplet radii) necessary for droplet transport in our system. Particularly, by reducing the magnitude of the chemical driving force[3], and accounting for the effect of roughness on the contact angle hysteresis force, we could articulate the effect of mechanically controlled microtextures on droplet transport behavior. This approach affords a mechano-responsive intelligent system capable of dynamic control of microdroplet transport: velocity, displacement, and critical radii of transport.

## Results
### Design and operation of mechano-responsive microdroplet transport devices.
To fabricate microtextured chemical gradients we combined a simple template-guided gas diffusion procedure[42] together with the stress-induced wrinkle formation process described above (Fig. 1a). We minimized the effects of stress-induced substrate curvature by fabricating 1 mm thick PDMS films and keeping the films flat during characterization. We then fully characterized the surface topography (Fig. 1b), roughness (Fig. 1c), and chemistry (Fig. 1d and Supplementary Fig. 1), of these microtextured chemical gradients at various states of compressive strain ($\varepsilon_c$) where $\varepsilon_c = \varepsilon_{pre} - \varepsilon$ and $\varepsilon_{pre}$ is pre-strain and $\varepsilon$ is strain. These topographic data were used to calculate the

Wenzel roughness factor ($r$, see Supplementary Discussion, section 1.3), which we expected to play an important role in droplet transport. We then investigated the motion of 3 µL water microdroplets deposited on the hydrophobic end of the chemical gradient at various compressive states (Fig. 1e). Water was used because it is a benign model solvent commonly employed in surface-fluidic technologies. A 3 µL droplet was transported 13.4 ± 0.8 mm with an average velocity of 4.3 ± 0.7 mm s$^{-1}$ at $\varepsilon_c = 0$ across the chemical gradient before its motion stopped (Fig. 1e, Supplementary Fig. 2, Supplementary Movie 1). The velocity of the deposited microdroplets gradually decreased as $\varepsilon_c$ increased, until $\varepsilon_c = 0.15$ was reached, where microdroplet transport ceased. This observation can be attributed (as discussed in detail below) to the increase in the corresponding surface roughness (Fig. 1c, Supplementary Fig. 3, Supplementary Movie 2), which we measured to be 1.00 ± 0.01 at $\varepsilon_c = 0$ and 1.15 ± 0.03 at $\varepsilon_c = 0.15$ (Fig. 1c). We investigated the reproducibility of this general observation by stress cycling a film between $\varepsilon_c = 0$ and $\varepsilon_c = 0.20$ observing no significant difference in droplet transport behavior up to 100 cycles (Supplementary Fig. 4). Specifically, microdroplet velocity on the film with $r = 1$ ($\varepsilon_c = 0$) remained constant ($v = 4.3 ± 0.7$ mm s$^{-1}$ after 0 cycles and $v = 4.3 ± 0.3$ mm s$^{-1}$ after 100 cycles, Supplementary Fig. 4). Further, the distance traveled by the microdroplets gradually decreased as $\varepsilon_c$ increased, providing a means to control the microdroplet's final resting position (Supplementary Fig. 5). It is possible to extend the droplet transport distance by tuning the gradient steepness and length[12]. Furthermore, we believe with additional development, periodic gradients could be used to extend transport beyond that possible using a single gradient.

### Physical model predictive of microdroplet transport characteristics.
We next sought to predict the condition where the resistive (negative) force was greater than the gradient driving force and microdroplet transport was not possible. Deterministic understanding of the system in this way would be necessary for the design and fabrication of sophisticated liquid transport devices with predictable performance characteristics. To do this we systematically probed the interplay between surface roughness, compressive strain, and microdroplet velocity to realize an expression—one based on empirical data that is specific to the system under investigation—capable of predicting the conditions necessary for transport.

We began by determining the "critical" radius ($R_c$)—the droplet radius where the velocity and thus the net force acting on the microdroplets was approximately zero ($F = 0$) and transport does not occur—by depositing a series of increasingly smaller droplets on a smooth gradient ($r = 1$, $\varepsilon_c = 0$) until transport ceased (Fig. 2a). We deposited the droplets at the same position on the hydrophobic end of the chemical gradient to mitigate position-dependent variations in the chemical gradient and measured $R_c$ along the axis of strain[3]. Following this approach, we measured $R_c$ to be 0.91 ± 0.04 mm (Fig. 2a), which corresponded to a critical volume ($V_c$) of ~1 µL. We then calculated $R_c$ using Eq. (4) (by setting $F = 0$, $r = 1$, and using measured/tabulated physical parameters), obtaining a value of 0.30 ± 0.04 mm, which was significantly lower than the measured quantity of 0.91 ± 0.04 mm (Fig. 2a). This difference (0.61 ± 0.06 mm) was indicative of either an additional negative force not accounted for by Eq. (4) or an overestimation of the driving force[3]. We believed overestimation of the driving force was likely, as supported below.

Several reports indicate that the difference between the driving force and hysteresis force (Eq. (4)) is not sufficient in describing droplet transport behavior on chemical gradients[3,38,40]. The

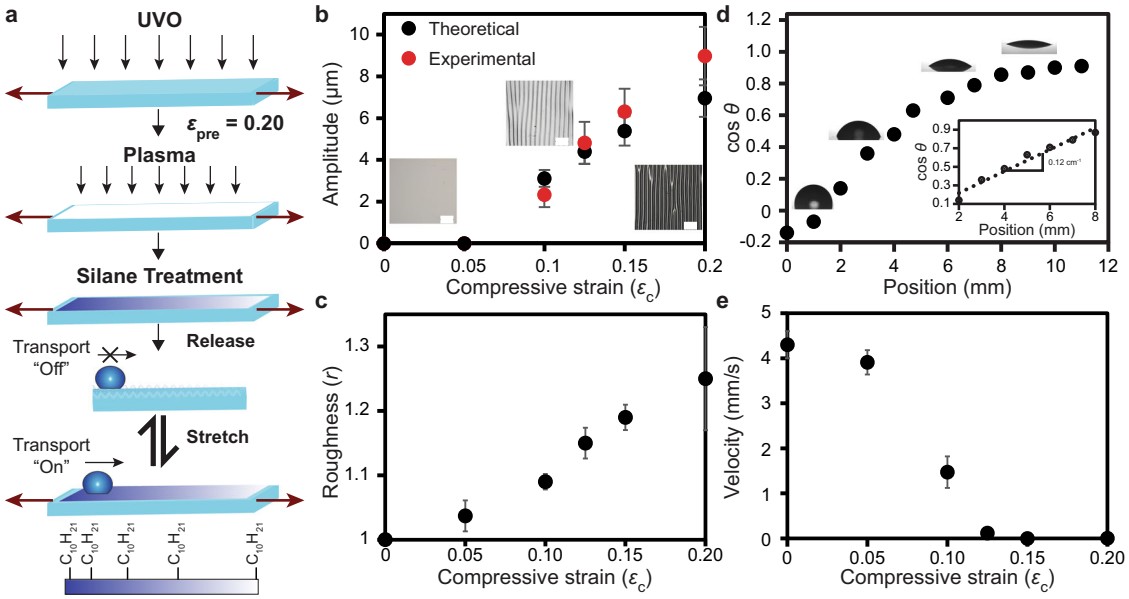

**Fig. 1 Fabrication and mechanically controlled droplet transport. a** Schematic illustrating the process for fabricating chemical gradients on films with mechanically tunable surface microtextures (wrinkles) and their use in mechano-activated droplet transport. **b** The theoretical and experimentally measured wrinkle amplitudes. Insets: optical micrographs of the surface topography at $\varepsilon_c = 0$, 0.10, and 0.20 respectively, (scale bars = 200 μm). ($N = 30$ for each data point, data reported as $\bar{x} \pm s$ where $\bar{x}$ is the mean and $s$ is the standard deviation). **c** Surface roughness of the microtextured gradients at increasing magnitudes of compressive strain ($\varepsilon_c$). ($N = 5$ for each data point, data reported as $\bar{x} \pm s$ where $\bar{x}$ is the mean and $s$ is the standard deviation). **d** Variation in the measured contact angle (θ) as a function of position for an uncompressed PDMS film. Inset: The contact angle variation for the linear portion of the gradient. **e** Transport velocity of 3 μL $H_2O$ droplets on the microtextured gradients at increasing magnitudes of compressive strain. ($N = 5$ for each data point, data reported as $\bar{x} \pm s$ where $\bar{x}$ is the mean and $s$ is the standard deviation).

prevailing explanation for this difference is that most studies overestimate the driving force (a systematic problem that could arise from differences in conditions for gradient characterization and transport studies)[3]. In this case, we expected that the difference in the predicted force ($F$, Eq. (4)) for a series of measured $R_c$ values (at variable roughness) would scale as $F \propto R_c^2$. To assess this expectation, we determined $R_c$ at compression states of $\varepsilon_c = 0$, 0.100, 0.125, 0.150, 0.1625 and 0.175. In order to minimize the effects of droplet anisotropy due to the wrinkled topography, $R_c$ was measured exclusively at the hydrophobic end of the chemical gradient where anisotropy was nominal. We proceeded to calculate $F$ (using Eq. (4) and experimentally measured parameters) for each compression state, and plotted $F$ versus $R_c$, $rR_c$, and $R_c^2$ (Supplementary Fig. 6, Supplementary Fig. 7, and Fig. 2b, respectively). We observed a better correlation between $F$ and $R_c^2$ (correlation coefficient $r_p^2 = 0.9974$, Fig. 2b) than between $F$ and $rR_c$ ($r_p^2 = 0.9759$, Supplementary Fig. 6) or $F$ and $R_c$ ($r_p^2 = 0.9705$, Supplementary Fig. 7), and verified that the differences between the $F$ versus $R_c^2$ correlation and the other two correlations were statistically significant using a $Z$-test ($P < 0.05$). We viewed the significant correlation of $F$ and $R_c^2$ together with the results of the $Z$-tests as strong support for the assertion that the difference in force was related to an overestimation of the driving force (which scales with $R_c^2$) and not the hysteresis term (which would scale with $R_c$), and thus applied a fitting parameter to the positive driving force term. It has also been proposed that an additional three phase contact line term is missing from Eq. (4), but this term should scale with $R_c$[37]. Worth mention is the effect of droplet adhesion, which would also scale with $R_c^2$[45]; however, droplet adhesion does not apply during motion along a surface[46,47].

To correct the magnitude of $F_d$, we used the calculated force value at $r = 1$ from Eq. (4) ($15.3 \pm 1.6$ μN) to determine the

magnitude of reduction in $F_d$ required to give $F = 0$ (Fig. 2a). Specifically, we introduced a fitting constant "C" into Eq. (4), where $C = 3.04$. We now express the modified force equation as:

$$F' = \frac{\pi}{C} R^2 \gamma \Phi - 2\gamma r R b \qquad (7)$$

To apply Eq. (7) to a range of compression states (not just those explicitly investigated), we expressed contact angle hysteresis ($b$ in Eq. (7)) as a function of $R$. To do so, we measured contact angle hysteresis by depositing a small range of water microdroplets ($R = 0.91$ mm to $R = 1.40$ mm) on the hydrophobic end of a smooth ($r = 1$) gradient surface, obtaining a linear correlation (Supplementary Fig. 8) over this range. We then substituted the resulting linear regression equation for $b$ into Eq. (7) for the purpose of predicting $R_c$. We acknowledge that this approach simplifies the complex behavior of contact angle hysteresis (which generally depends on a number of factors including surface chemistry homogeneity, topography, and the size of liquid droplets)[35,48–50], but believed it sufficient given the linear correlation over the range of radii examined in our studies.

Using this approach, we applied Eq. (7) to the smooth chemical gradient data set (Fig. 2a), yielding good agreement between the calculation and empirical observation (predicted $R_c = 0.84$ versus measured $R_c = 0.91 \pm 0.04$ mm) (red trace, Fig. 2a). Further, we used Eq. (7) to predict critical droplet radii of 0.97 mm, 1.19 mm, 1.59 mm, 1.77 mm, and 1.98 mm at $\varepsilon_c$ values of 0.10, 0.125, 0.150, 0.1625, and 0.175, respectively (Fig. 2b, c). These predicted values are statistically indifferent ($P < 0.06$) from experimental observations of $1.09 \pm 0.06$ mm, $1.22 \pm 0.03$ mm, $1.54 \pm 0.06$ mm, $1.77 \pm 0.03$ mm, and $2.13 \pm 0.10$ mm, respectively (Fig. 2c). Furthermore, we show that when roughness is not included in Eq. (7) ($F''$, blue trace, Fig. 2c), the calculated $R_c$ does not match experimental observations.

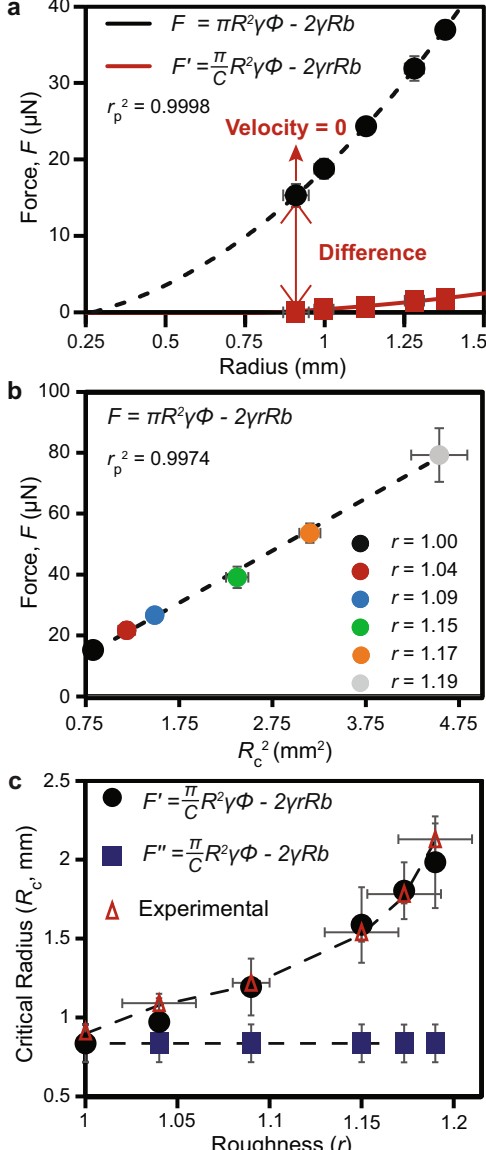

**Fig. 2 Force balance and the relationship between Critical Radius and Roughness. a** Microdroplet force analysis on a smooth surface gradient highlighting the difference between equation 4 (black data points) and experimental observations ($F'$, red data points). Both data sets were fitted to a second order polynomial (black and red traces respectively). **b** Force differences plotted against $R_c^2$ (Pearson's correlation coefficient of the linear regression is denoted as $r_p^2$). $N = 5$ for each data point, data reported as $\bar{x} \pm s$ where $\bar{x}$ is the mean and $s$ is the standard deviation. The error in force measurements were obtained following the propagation of uncertainty using equation (4). **c** Comparison of the predicted and experimental critical radii ($R_c$) at various roughness states ($r$). Shown are the reduced driving force model ($F'$, Eq. (7)), the reduced driving force model without roughness ($F''$), and the experimental results.

**Mechano-adaptive microdroplet transport**. We utilized our understandings of microdroplet transport on chemical gradients with mechanically controlled microtextures to demonstrate dynamic surface-fluidic capabilities and, by extension, expand the practical capabilities of gradient-based droplet transport (Fig. 3). The most fundamental capability stemming from this concept is switching transport "on" from an "off" state, using the appropriate modulation of $\varepsilon_c$ (Fig. 3a). To demonstrate this critical capability, we set $\varepsilon_c$ to 0.20 (the condition where microdroplets do not move)

and, in a single step, decreased $\varepsilon_c$ to 0.10 (Fig. 3b and Supplementary Movie 3). As expected, a 3 μL microdroplet remained stationary on the surface until $\varepsilon_c$ was decreased (at 5.85 s), at which point microdroplet transport commenced, reaching a peak velocity of 21.7 mm s$^{-1}$ (Fig. 3c). This change in $\varepsilon_c$ represents a decrease in roughness from $1.25 \pm 0.08$ to $1.04 \pm 0.02$, which meets the condition described by Eq. (7) for a positive net force acting on the droplet.

Extension of this capability enables programmable and reversible switching of the surface between transport "on" and "off" states multiple times (Fig. 3d). To demonstrate this capability, we deposited a microdroplet on the film at $\varepsilon_c = 0.10$, a condition where transport was allowed, and rapidly increased $\varepsilon_c$ to 0.15 after 1.05 s, immediately suspending microdroplet transport (Fig. 3e and Supplementary Movie 4). We then switched back to $\varepsilon_c = 0.10$ at 5.35 s, enabling microdroplet transport to continue. It was again possible to toggle transport "off" then "on" once more by increasing $\varepsilon_c$ to 0.15 and then decreasing $\varepsilon_c$ to 0.05 (Fig. 3e and Supplementary Movie 4). The velocity profile clearly highlights the controlled modulation of fluid transport repeatedly along a single gradient surface (Fig. 3f). Further analysis of the dynamic transport experiments above demonstrates that the velocity of microdroplet transport is also variable and controllable using $\varepsilon_c$ (Fig. 1e, Fig. 3c, and Fig. 3f).

Previously, chemical gradients were used to transport water droplets against gravity (up a 15° incline)[1]. We are able to elevate this functionality using our system by rationally transporting microdroplets up (or down) inclines in an "on demand" fashion (Fig. 4). Specifically, we demonstrated the ability to modulate microdroplet motion along a 30° inclined plane (Fig. 4a). A 3 μL microdroplet was deposited on the hydrophobic end of the inclined film with $\varepsilon_c$ set to 0.20, the condition where transport is not possible (Supplementary Movie 5). We then stepped $\varepsilon_c$ to 0.10 and droplet transport proceeded as shown by the rapid increase in its velocity (Fig. 4b, c and Supplementary Movie 5). This mechano-switching of droplet motion at a 30° incline could be modulated multiple times, highlighting the dynamic control of transport accessible using this approach (Fig. 4b, c and Supplementary Movie 5).

Similarly, we highlighted the ability for dynamic surface microtopography to prevent a microdroplet from freely moving down the inclined film by controlling the surface microtopography (and therefore increasing the hysteresis force to a magnitude greater than that of the summative driving force from the gradient and gravity) (Fig. 4d). Once again, in the presence of wrinkles at $\varepsilon_c = 0.20$, transport was not possible, and the droplet remained pinned despite the drive of the gradient and gravity (Fig. 4e, Supplementary Movie 6). When we decreased $\varepsilon_c$ to 0.10, transport proceeded and the microdroplet was accelerated along the downward slope (Fig. 4e, f), further emphasizing the flexibility of this approach for controlling microdroplet transport across a range of substrate orientations.

**Applications of mechano-adaptive microdroplet transport systems**. In addition to modulating droplet transport and velocity along horizontal and inclined planes, these films were used as mechano-activated self-cleaning devices (Fig. 5a). First, we fabricated a film with two repeating chemical gradients, and "dirtied" the film by covering the surface with metal dust (which provides convenient optical contrast). Subsequently, we deposited water microdroplets on the hydrophobic end of each chemical gradient at $\varepsilon_c = 0.20$ where microdroplet transport was not expected (Fig. 5b). We then activated the self-cleaning process by decreasing $\varepsilon_c$ to 0 effectively reducing the surface roughness to 1. Surprisingly, even in the presence of surface particles—which is

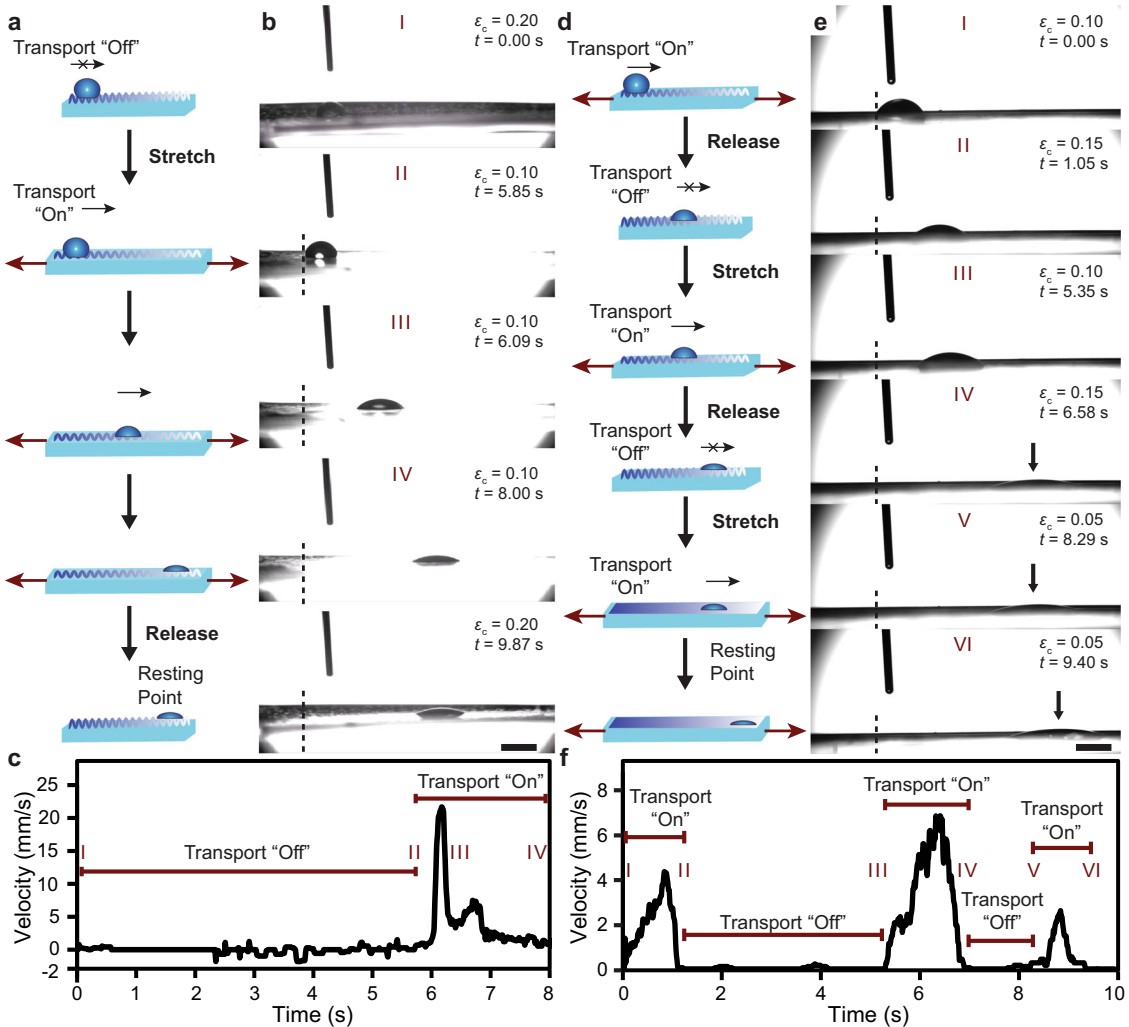

**Fig. 3 Controlling droplet transport dynamically using mechanical deformations. a** General schematic and **b** optical micrographs illustrating the dynamic transition of fluid transport from an "off" state to an "on" state using mechanical strain (compressive strain, $\varepsilon_c$ and time, $t$, are given). **c** Temporal velocity profile of the microdroplet extracted from the control sequence in **b**. **d** Schematic and **e** optical micrographs illustrating the dynamic toggling of fluid transport between "on" and "off" states using mechanical strain. **f** Temporal velocity profile of the microdroplet extracted from the control sequence in **e**. Scale bars = 2 mm.

known to alter surface topography/roughness—microdroplet transport still proceeded along the chemical gradient, removing all particles in its path, thereby cleaning the surface in this process (Fig. 5b, Supplementary Movie 7).

Finally, by taking advantage of the relationship between $R_c$ and $r$ (Fig. 2c), we developed a simple droplet sorting device (Fig. 5c, Supplementary Movie 8). As demonstrated, when roughness increased, the critical radius also increased, and therefore larger volume droplets were required for transport on rougher surface gradients (Eq. (7), Fig. 2c). Thus, a droplet with a larger radius (volume) was expected to move faster (Eq. (6)) and prior to a droplet with a smaller radius (Eq. (7)) as roughness decreased (Fig. 5c). We highlighted this functionality by depositing three microdroplets of increasing volumes (1 μL, 2.5 μL, and 5 μL) on the hydrophobic end of the chemical gradient at $\varepsilon_c = 0.20$. As expected, droplet motion was not observed at this compressive strain (Fig. 5d). We then reduced the roughness by decreasing $\varepsilon_c$ to 0.10 ($r = 1.04$), initiating transport of the largest (5 μL) droplet across the gradient as the net force balance (Eq. (7)) became positive (Fig. 5d). Under this condition the net force balance (Eq. (7)) for the smaller 2.5 μL and 1 μL droplets remained negative and they did not move. We then initiated transport of the 2.5 μL microdroplet by reducing $\varepsilon_c$

further to 0 ($r = 1$) where the force balance (Eq. (7)) became positive for this droplet (Fig. 5d). The 1 μL droplet was below the critical radius required for transport under any roughness condition and remained at the point where it was initially deposited. This simple droplet sorting device accentuates the utility of the relationship between the critical droplet radius and roughness as it pertains to constructing fluid transport devices of increasing sophistication using microtextured surface-chemical gradients.

**Discussion**

We demonstrated a simple yet powerful approach for the dynamic handling and transport of liquid microdroplets using chemical gradients with mechanically switchable microtextures. In addition, we developed a predictive model—one that accounts for a reduction of the driving force and surface roughness—to determine the conditions (microdroplet radius and surface roughness) required for programmable droplet transport. Using these understandings, we demonstrated the ability to control microdroplet velocity and displacement reproducibly using mechanical strain and highlighted the on-demand reversible programming of microdroplet transport on horizontal and tilted surfaces. We further showcased the usefulness of these films in

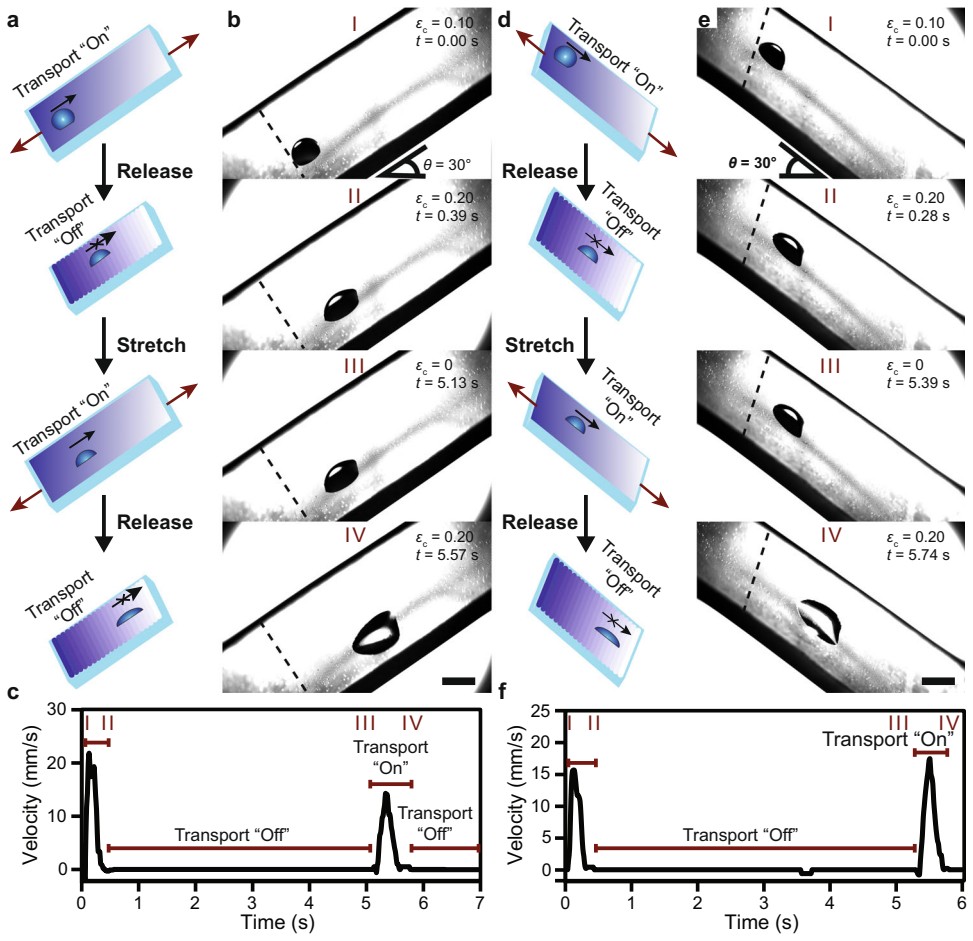

**Fig. 4 Controlling droplet transport dynamically along inclined surfaces using mechanical deformations. a** General schematic and **b** optical micrographs demonstrating dynamic toggling of fluid transport between "on" and "off" states on a 30° inclined plane. **c** Temporal velocity profile of the microdroplet extracted from the control sequence in **b**. **d** General schematic and **e** optical micrographs demonstrating dynamic toggling of fluid transport between "on" and "off" states on a 30° declined plane. **f** Temporal velocity profile of the microdroplet extracted from the control sequence in **e**. Scale bars = 2 mm.

mechanically activated self-cleaning materials and volume-based droplet sorting devices. Despite the simplicity of fabrication and operation, technologies based on microtextured chemical gradients have the potential for relatively sophisticated functionality due to the large number of control parameters (e.g., $\varepsilon_{pre}$, $\varepsilon_c$, gradient chemistry and geometry, microdroplet composition and size, etc.) which enable optimization of the transport properties.

We have focused on one PDMS formulation, a specific set of oxidation parameters and pre-strain, and a well-studied type of chemical gradient, but it will be possible to access different microtextures with distinct roughness states and chemistries using different fabrication conditions and/or procedures. The general concepts described in this work are applicable to other elastomers which may provide a range of mechanical and/or chemical properties not possible using PDMS. In designing these protocols and/or selecting new materials, it will be necessary to avoid surface cracking and other irreversible surface microstructural changes, which we achieved here through carful optimization of the oxidation conditions. Furthermore, the magnitude of the driving force deviation from ideal values is system dependent and the stability of the surface-chemical gradients (which is typically on the order of days) must be considered for specific applications and appropriate storage measures taken when necessary[51].

We aimed to demonstrate/characterize the fundamental aspects of dynamic droplet transport across single, one-dimensional (linear) gradients, which are ideally suited for water harvesting, anti-fouling, and self-cleaning applications. Further advancement of our system (i.e., incorporation of periodic gradients and/or energy input to aid in the transition of droplets across the surface energy barrier present at gradient boundaries) will be necessary to realize the full potential of this technology. We believe that extensions of the demonstrated concepts will lead to more sophisticated surface-fluidic capabilities useful to, for example, biomedical and analytical devices, mechano-switchable water sorting devices, surface lab-on-chip devices, and adaptive materials for emergent robotic applications. For example, the use of mechano-activated, two-dimensional (non-linear), periodic gradients together with the input of additional kinetic energy (i.e., energy from mechanical vibration) could enable advanced lab-on-chip functionality. In this way the inherent "one-way" transport of chemical gradients could be overcome, and more intricate droplet manipulations realized. We anticipate that the core capabilities reported here which leverage mechanically tunable, dynamic, microtextured surface-chemical gradients represent a critical first step in synthesizing programmable droplet transport systems with increasingly advanced functionalities.

## Methods

**Materials**. The silicone elastomer used was PDMS, Sylgard® 184 silicone elastomer kit (Dow Corning, Midland, MI). Acrylonitrile butadiene styrene (ABS plus) model

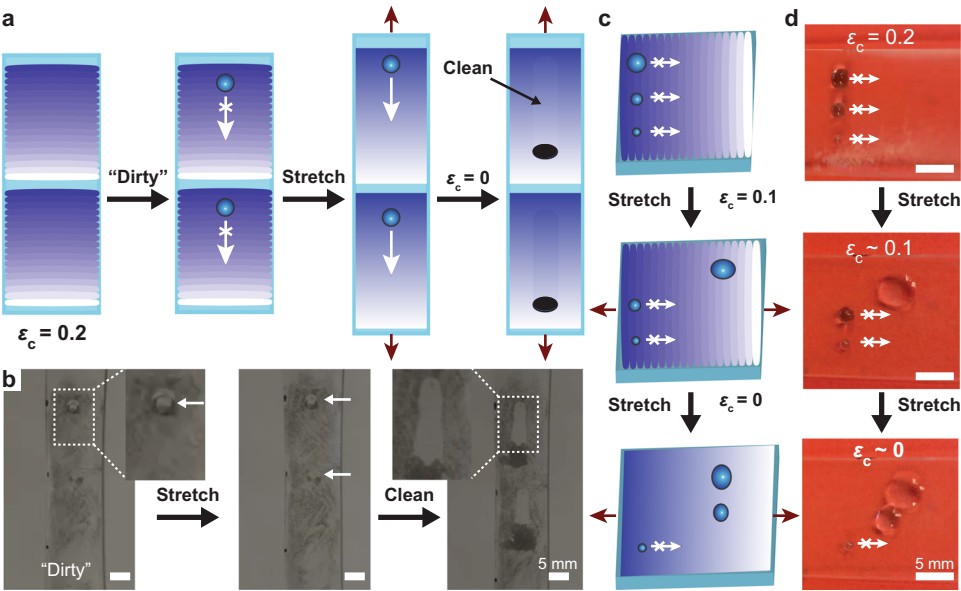

**Fig. 5 Mechanically activated self-cleaning surfaces and mechanically controlled droplet sorting devices. a** Schematic illustration of the use of mechanical deformations to activate droplet transport and thus self-cleaning functionality. **b** Optical micrographs of the mechanically activated self-cleaning process (insets are higher magnification of the highlighted regions, and compressive strain, $\varepsilon_c$ is given). **c** Schematic illustration of the operation of a mechanically controlled droplet sorting device. **d** Optical micrographs of a droplet sorting device in operation. Transport of a 5 μL droplet occurs following activation of the gradient using a compressive strain of $\varepsilon_c = 0.1$. Transport of a 2.5 μL droplet occurs following maximal activation of the gradient by reducing compressive strain to $\varepsilon_c = 0.0$. The 1 μL droplet is below the critical radius of transport for this device and does not move.

material (Stratasys Ltd.) was used for 3D printing. For the fabrication of chemical gradients, decyltrichlorosilane (TCI America, Portland, OR) was diluted in light mineral oil (Fisher Scientific, Chicago, IL), and both were used as received. Iron powder (Fisher Scientific, Chicago, IL) was used as received for self-cleaning experiments.

**Fabrication of PDMS films**. PDMS films were prepared following manufacturer protocol. Briefly, a mixture of 10:1 base to curing agent prepolymer was degassed in a vacuum desiccator for 20 min. The degassed PDMS was poured inside a 3D-printed mold resting on top of a pre-cleaned silicon wafer (to ensure 1 mm film height). Then the prepolymer mixture was cured inside a convection oven at 60 °C for 24 h.

**Oxidation of PDMS films**. PDMS was cut into $4 \times 2$ cm films and strained to $\varepsilon_{pre} = 0.20$. Then the pre-strained films were place under an UVO lamp (Grid Lamp, AssY, 5" × 5" × 0.5" BHK Inc., Claremont, CA) with 3 mm separation between the films and the lamp for 90 mins. Afterwards, the films were briefly oxidized inside an $O_2$ plasma oxidation chamber (Plasma Etch Inc., Carson City, NV, Model #PE-25 Series) at 15 W power for 5 s to render the surface hydrophilic.

**Wrinkle generation and characterization**. Mechanical pre-strain was released leading to spontaneous wrinkle formation. The surface microtopography of the wrinkled PDMS films was characterized using confocal microscopy (Confocal Microscope Keyence Variance, VK-X200Series). Specifically, we used confocal microscopy to measure the critical strain ($\varepsilon_{cs}$)—the strain where wrinkles begin to form —wrinkle pitch ($\lambda$), wrinkle amplitude ($A$) at variable strain, and the Wenzel's roughness factor ($r$). We systematically compared these values to established mechanical models in order to validate the idealized behavior of our system and to guide design/predictability of the topographic features of our microtextured surfaces.

**Chemical gradient synthesis**. We fabricated chemical gradients on the surface of the oxidized films following a procedure developed in our lab[42]. Briefly, we created a 1:10 decyltrichlorosilane (DTCS): light mineral oil solution. We then soaked precut filter paper with 15 μL of this solution to decrease the volatility. The filter paper was then inserted into a 3D-printed diffusion chamber, which was then placed on top of the oxidized films in a sealed environment to allow for controlled diffusion and surface functionalization. Afterwards, the chamber was removed, and the films were immersed in a 60 °C DI water bath for 60 s. Finally, the films were rinsed with aliquots of DI water for 30 s.

**Contact angle and gradient intensity measurements**. We obtained side view optical images using an Attension Theta contact angle goniometer (Biolin Scientific, Gothenburg, Sweden) to quantify the gradient intensity $\Phi$. Specifically, we deposited 0.20–0.30 μL DI H$_2$O microdroplets continuously along the chemical gradient at 1 mm spacing intervals, where contact angle measurements were

repeated in triplicates for each individual data point. We constructed a plot of cos θ relative to the gradient position. We then used the linear portion of the plot to calculate $\Phi$.

**Contact angle hysteresis**. We measured the difference between the advancing and receding contact angles for droplets of various radii using the Attension Theta contact angle goniometer. Due to probing incipient motion, we measured contact angle hysteresis near the onset of motion—the condition most relevant to our study.

**Measuring transport velocity**. Microdroplets of differing radii were deposited on the surface of chemically modified films. We varied $\varepsilon_c$, incline/decline angles accordingly, and recorded movies using the Attension Theta contact angle goniometer's high-speed camera. Movies were edited using the Premiere Pro software. Instantaneous velocity data were determined from these movies by extracting a down sampled series of frames (down sampled to 69 frames per second) for position/time analysis using the open-source software package ImageJ. Specifically, for each frame series, the center of a 3 μL DI H$_2$O droplet was tracked during the transport sequence using the MTrack2 plugin. The magnitude of the pixel change following each successive frame was measured and converted to its corresponding distance (mm). We used a 10-frame moving average in order to reduce the noise in the instantaneous velocity measurements.

**Mechanically responsive microdroplet transport**. We used a custom-made electromechanical tensioning device operated using open source Adruino$^{TM}$ microcontrollers, commercial stepper motors, and a custom GUI written using Python$^{TM}$. We set the rate of strain to 30 mm s$^{-1}$ and varied the "hold at strain" value and duration based on the desired control sequence. In addition, we used an Instron Materials Testing System (Norwood, MA, model #5944) for mechanical stress cycling experiments and demonstrating the self-cleaning capabilities of our system. Here, the rate of strain was set to 500 mm min$^{-1}$ and the hold at strain step was set to 10 s.

**Self-cleaning films**. We fabricated two consecutive chemical gradients on the surface of a PDMS film (length = 6.0 cm) following the general described procedure. We then coated the surface with iron dust, attached the film to the Instron tensioning instrument, and carefully deposited a 3 μL DI water droplet on the hydrophobic ends of each gradient. Following droplet deposition, we strained the film to $\varepsilon_{pre}$ at which point the microdroplets began traversing across the chemical gradients, cleaning the surface in the process.

**Volume sorting device**. We fabricated a microtextured chemical gradient following the general outlined procedure. We carefully deposited three blue dyed microdroplets

(to enhance contrast) with increasing volumes (1 μL, 2.5 μL, and 5 μL) on the hydrophobic end of the chemical gradient. We then slowly decreased compressive strain to $\varepsilon_c = 0.1$ using a mechanical tensioning device causing the 5 μL droplet (largest volume/radius) to move across the gradient. We repeated the same process but decreased the compressive strain to $\varepsilon_c = 0$ at which point the 2.5 μL droplet moved across the gradient. The radius of the 1 μL droplet was approximately equivalent to the critical radius, therefore it did not move.

## Data availability

The data that support the findings of this study are available from the corresponding author upon reasonable request.

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

## Acknowledgements

We thank the Department of Chemistry and the Nebraska Center for Materials and Nano Science (NCMN) at the University of Nebraska–Lincoln for start-up funds. This work was supported by the National Science Foundation under Grant No. 1555356. This research was performed in part at the Nebraska Nanoscale Facility: National Nanotechnology Coordinated Infrastructure and NCMN, which are supported by the National Science Foundation under Award ECCS: 1542182, and the Nebraska Research Initiative.

## Author contributions

A.J.M, J.J.B, and S.A.M conceptualized the design of the materials and experiments. A.J.M., J.J.B., and J.M.T. conducted experimental investigations with A.J.M. playing the central role in method development and validation. A.J.M and S.A.M. wrote the paper with editorial revisions and input from J.J.B and J.M.T. S.A.M. managed the project as the principal investigator.

## Competing interests

The authors declare no competing interests.

**Additional information**

