## [Peer Review File · Nature Communications]

REVIEWER COMMENTS

Reviewer #1 (Remarks to the Author):

The authors describe a novel method to manipulate droplets based on the combination of chemical gradients and surface texture which is tunable by applying a mechanical stress. The results build on previous work from the group and are the combination of two techniques previously studied independently for droplet transport. The possibility of dynamically controlling the motion is great and I can see the potential for this technology when developed further. I believe the article should be eventually published in Nature Communications after taking into account some important remarks that I would like the authors to address. In particular,

1. Both chemical gradients and structural gradients on elastomer supports have been used separately by the group to induce droplet motion (Refs. 12 and 35). The authors should thoroughly discuss and evidence the advantage of using their combined approach in the dynamic control of droplets. This is not evident in the current narrative.

2. The literature on "programmable droplet transport", "controllable droplet manipulation", etc, on solid surfaces has expanded significantly in recent years with many novel methods appearing based on several principles, beyond electrowetting or SLIPs. The authors should make sure to cite the relevant literature on the topic. A few examples of these mechanisms include light-activated chemical gradients (Ichimura et al., Science 288, 1624–1626 (2000) and Berná et al., Nat. Mater. 4, 704–71 (2005)), surface-charge printing (Sun et al., Nat. Mater. 18, 936–941 (2019)), localised vapour sources (Malinowski et al., Science Advances 6, eaba3636 (2020), mechanowetting (De Jong et al., Science Advances 5, eaaw0914 (2019)), etc. Recent reviews on the topic also exists.

3. In the data and videos presented, the droplets only move 2-3 droplet's diameters. This distance is relatively short for dynamic control of droplets where one would like to achieve several droplets' diameters at least. Is it possible to extend the dynamic range of droplet control? Can the authors comment on this?

4. Similarly, compared to other manipulation approaches (see above), this method seems to be a one-way form of transport. While this might be enough for some applications (water harvesting, energy generation, self-cleaning surfaces), it is a limitation for other applications of droplet transport. In fact, in several videos, it can be seen that the droplets fully wet the surface at the end of its journey due to the high wettability of the surface and, thus, do not retain their shape. The authors should clarify this point in their manuscript.

5. The authors modify the general expression for the force on chemical gradients first used by Daniel et al. and subsequently modified by Shastry et al. to account for surface texture. They then introduce a fitting constant to explain the discrepancy between model prediction and experimental data. Although this discrepancy has been reported by other authors, the way in which the authors correct for it in their work is purely phenomenological. I would recommend clarifying this aspect in their manuscript

6. In their model, the authors modify the expression for the force that has a power 2 dependence on R to account for discrepancy between prediction and experiments. They justify based on fitting the discrepancy to trends depending on R^2 , rR and R . Based on the Pearson correlation, all trends can be plausible. The authors should try to fit over a larger range of R values in order to come to a more convincing conclusion, otherwise the choice of where to introduce the fitting parameter remains arbitrary. For example, the effect could also depend on an underestimation of the hysteresis force (Eq. 2), which is often reported with a prefactor that can take values up to n depending on the exact shape of the three-phase contact line. The authors touch briefly on the topic but the argument based on the previous fitting is not convincing.

7. Connected to the previous point, the authors introduce the fitting constant by renaming the contact angle gradient. I would separate the two things and just introduce a separated fitting constant as the contact angle gradient is a physically measurable parameter that depends on the

material and has been measured experimentally by the authors.

8. Both in main text and supplementaries, the authors mention that they neglect the viscous drag force. However, this force is very fundamental for the force balance in the motion of the droplet and it's used by the same authors to obtain an expression for velocity later in the text (Eq. 5). This should be clarified. I was also unclear on how the authors go from expression 3 to expression 5. This should be clarified, and a self-standing derivation should be included with all assumption and approximations made rather than being left to the supplementary information.

A few minor points:

9. On line 125, the authors mention a pre-strained PDMS film. Details on the pre-straining process should be provided.

10. Statistical information (number of experiments, type of error, etc) should be included in all figure captions.

11. Surfaces seem to show a curvature (probably due to the mechanical stress). Can this curvature influence the authors' results?

12. The quantities in the supplementary shouldn't be bold unless they are vectors.

Reviewer #2 (Remarks to the Author):

The authors combined the chemical gradient and surface wrinkle to manipulate the droplet transportation on the surface. They detailed the discussion about the driving force, resistant force and the moving velocity of the droplet. While the discussion about the driving velocity could be interesting to the field, the technical design and the results are somehow not impressed. The following are detailed comments.

1. The motivation of the current study is somehow unclear, the combination of several techniques cannot be reasoned as the motivation. I noticed that the references are rather old, and many important new works are missing. For instance, Nature Materials, 2019,18, 236, which can transport a droplet at a very large speed and to a long distance, and even on an upside down surface. Sci. Adv. 2017, 3, eaao3530 also reported a surface design for guiding the water transportation; Adv. Sci. 2020, 2001650 reported a surface presents a similar strategy using stress to regulate the surface pattern and then the state-switching of droplet on the surface. Actually, there are several more works making use of the stress.

2. How was the wrinkle prepared, if an unidirectional stress was used, then the wrinkles have a certain orientation, it will have a strong influence on the shape of the droplet on it, then, how was the radius of the droplet determined? From the video, we can see that many droplets are extremely asymmetric. Due to the fabrication process, any anisotropic properties of the droplets? Like the contact angle, and also the transportation behaviors?

3. P5, Quantitative characterization about the surfaces, like the structure parameters, surface energy and the gradients should be given in the main text, which are very important for the following discussions.

4. How is the state of the droplet on the surface, it is in Cassie or Wenzel state? Both cases fit to the model proposed?

5. Fig2a, the authors discussed the critical size of the droplets, however, without the precise description of the surface, it is meaningless. When a larger droplet is placed on the surface, surely, it will have a larger driving force as a larger contact area and thus a larger gradient exists.

6. "The most fundamental capability stemming from this concept is switching transport "on" from an "off" state", however, this kind of switching has been widely reported. like Adv. Sci. 2020, 2001650; Adv. Mater. 2017, 29; Adv. Funct. Mater. 2018, 28, 1800625 et al. Is the switching between two states reversible?

Title: “Dynamic Manipulation of Droplets using Mechanically Tunable Microtextured Chemical Gradients”

Authors: Ali J. Mazaltarim, John J. Bowen, Jay M. Taylor, and Stephen A. Morin*

Institution: University of Nebraska–Lincoln

Figures: 5

Dear Referee 1 and Referee 2,

We have revised the manuscript and supporting information to address your comments and recommendations. This revision includes substantial modifications to the introduction and discussion sections, the inclusion of additional critical radii data and related details concerning the numerical fitting parameter applied to the driving force term, a more thorough discussion of the relevant droplet transport equations, and a substantial revision of Fig. 1 to include critical surface characterization data. We have also added an additional 6 references in response to your comments. We include detailed responses to each of your individual comments starting on page 2. To assist in your review, we have included a clean version of the revised manuscript and a version with changes tracked in red text.

Sincerely,
Stephen A. Morin
Associate Professor
Department of Chemistry
University of Nebraska – Lincoln

Referee 1:

Note: The page and line numbers below refer to the clean version of the manuscript and S.I., not the marked versions.

From the Overview:

“The authors describe a novel method to manipulate droplets based on the combination of chemical gradients and surface texture which is tunable by applying a mechanical stress. The results build on previous work from the group and are the combination of two techniques previously studied independently for droplet transport. The possibility of dynamically controlling the motion is great and I can see the potential for this technology when developed further. I believe the article should be eventually published in Nature Communications after taking into account some important remarks that I would like the authors to address. In particular,”

We agree with the referee’s general sense for the potential of this technology and view this manuscript, which focuses on the fundamental underpinnings and basic capabilities of this approach, as an essential first step toward its future development. We have carefully considered all the referee’s thoughtful comments, responding in detail below.

Numbered Comments:

- 1) *“Both chemical gradients and structural gradients on elastomer supports have been used separately by the group to induce droplet motion (Refs. 12 and 35). The authors should thoroughly discuss and evidence the advantage of using their combined approach in the dynamic control of droplets. This is not evident in the current narrative.”*

We approached the challenge of dynamic droplet transport having valuable experience in this area from our work with both chemical and structural gradients on soft materials, as noted by the referee. From our perspective, chemical and microstructural approaches to droplet transport are limited mainly in terms of rapid switchability—to switch a chemical gradient, subsequent processing (surface-chemical modifications) are required, which can be slow and difficult to reverse; to switch a microstructural gradient, modifications to the highly specific geometric parameters of the wettability patterns are necessary, which can be challenging given the complicated nature of the microstructure-transport relationship. Our approach removes the requirement for additional chemical processing and highly refined modifications to microstructural patterns simultaneously, providing a new strategy for programmable droplet transport based on simple mechanical input. Here, we controlled droplet trajectory using the direction/profile of the chemical gradients, which, relative to structural gradients that require multi-step lithographic fabrication procedures, are easy to synthesize. We then tuned the effective “action” of the gradient using mechano-tunable microwrinkles, which are simple to fabricate and operate (the relationship between strain and structure is easily established) and defect tolerant compared to structural gradients. We are confident that this combined approach represents a substantial advance in capabilities inaccessible to either chemical or structural gradients acting alone. We have significantly amended the introduction to articulate these important ideas.

In addition to the above strategic advantages, there are also pragmatic advantages to combining chemical and microstructural features in dynamic transport systems. For example, unlike electrostatic gradients, our system is not susceptible to environmental effects (i.e., humidity).

Furthermore, our system can transport numerous droplets in sequence whereas electrostatic gradients must constantly be regenerated for droplet transport to occur. Lastly, once the films described in our work have been fabricated, dynamic transport can, in principle, be achieved using mechanical stimulation from various sources (e.g., electromechanical devices, user biomechanics) unlike systems requiring specific chemical, optical, or vibrational inputs.

We have added the following to the main text:

Beginning on page 2, line 10: “Accordingly, recent developments have leveraged fundamental understandings of surface microdroplet transport¹³ in an effort to realize new strategies for the design and fabrication of devices capable of increasingly dynamic and sophisticated manipulations of droplets^{4,5,8,13–17}. Despite these efforts, dynamic transport technologies require further evolution to overcome key limitations: switching between transport states using chemical/structural gradients can be slow and difficult to accomplish (e.g., photoactivated surfaces are tied to the kinetics of surface reactions and structural gradients have highly specific transport/geometry relationships)^{4,5,9–12}, continuous transport using electrostatic systems requires additional processing and environmental conditions can impact performance⁸, and in many cases, a dedicated input system (e.g., electrical power supplies, light sources, or vibrational stages) is required to stimulate/control transport^{2,4,5,13,16}.

...that did not require additional processing to control/operate once fabricated. Our new strategy, which combines the action of chemical gradients with mechanically tunable surface microstructure, simultaneously removes the requirement for additional chemical processing and highly refined modifications to microstructural patterns, providing a new strategy for programmable droplet transport based on simple mechanical input. This approach significantly elevates the practical applications of programmable droplet transport systems, and yields environmentally stable systems capable of rapid/instantaneous switching of droplet transport following mechanical stimulation from a potentially diverse range of sources (e.g., biomechanical input, electromechanical systems, or environmental stressors). We believe our versatile approach, which is devoid of static/rigid components, represents an important advancement in adaptive/programmable droplet transport capabilities and will have relevant applications in soft materials (i.e., wearables), adaptive systems (i.e., soft robotics), water-harvesting technologies, and energy generation.”

- 2) *“The literature on “programmable droplet transport”, “controllable droplet manipulation”, etc., on solid surfaces has expanded significantly in recent years with many novel methods appearing based on several principles, beyond electrowetting or SLIPs. The authors should make sure to cite the relevant literature on the topic. A few examples of these mechanisms include light-activated chemical gradients (Ichimura et al., Science 288, 1624–1626 (2000) and Berná et al., Nat. Mater. 4, 704–71 (2005)), surface-charge printing (Sun et al., Nat. Mater. 18, 936–941 (2019)), localised vapour sources (Malinowski et al., Science Advances 6, eaba3636 (2020), mechanowetting (De Jong et al., Science Advances 5, eaaw0914 (2019)), etc. Recent reviews on the topic also exists.”*

We have added the following references in the main text as suggested. We have also added a recent review on programmable droplet transport by Malinowski et al. (Ref. 13). Where necessary we added additional detail to provide context to the new references.

Page 2, line 5, (references 4, 5): “chemical¹⁻⁵...”

Page 2, lines 5-6 (reference 8): “surface charge density⁸,...”

Beginning on page 2, line 10 (references 4, 5, 8, 13, 16): “Accordingly, recent ... and energy generation.”

4. Ichimura, K., Sang-Keun, O. & Nakagawa, M. Light-Driven Motion of Liquids on a Photoresponsive Surface. *Science* **288**, 1624–1626 (2000).
 5. Berná, J. *et al.* Macroscopic transport by synthetic molecular machines. *Nature Mater.* **4**, 704–710 (2005).
 8. Sun, Q. *et al.* Surface charge printing for programmed droplet transport. *Nat. Mater.* **18**, 936–941 (2019).
 13. Malinowski, R., Parkin, I. P., Volpe, G., Advances towards programmable droplet transport on solid surfaces and its applications. *Chem. Soc. Rev.* **49**, 7879-7892, (2020).
 16. De Jong, E., Wang, Y., Den Toonder, J. & Onck, P. R. Climbing droplets driven by mechanowetting on transverse waves. *Sci. Adv.* **5**, eaaw0914 (2019).
- 3) “*In the data and videos presented, the droplets only move 2-3 droplet's diameters. This distance is relatively short for dynamic control of droplets where one would like to achieve several droplets' diameters at least. Is it possible to extend the dynamic range of droplet control? Can the authors comment on this?*”

In the current work, droplet transport can be viewed as a discrete translation step which is fundamentally limited by the length of the chemical gradient. We have worked extensively in the development of techniques which provide tunable, large-area, periodic chemical gradients (Ref. 18). Here, with simple linear gradients, we measured a maximum droplet transport distance of 13.4 mm ($R = 1.5$ mm and $r = 1.00$), or ~ 5 droplet diameters, though our goal was not to maximize distance. Using our ability to program the gradient shape and steepness it would be possible to further optimize the gradients and extend transport range. The intricate force balance acting on the droplet dictates that such modifications will result in predictable performance tradeoffs (e.g., a longer gradient would result in a lower velocity). We believe it will be possible to further extend transport range beyond that possible using a single gradient by combining periodic gradients with additional stimuli (e.g., mechanical vibrations).

We have added to and revised the main text as follows:

Page 7, lines 13-18: “A 3 μL droplet was transported 13.4 ± 0.8 mm with an average velocity of 4.3 ± 0.7 $\text{mm}\cdot\text{s}^{-1}$ at $\varepsilon_c = 0$ across the chemical gradient before its motion stopped (Fig. 1e, Supplementary Fig. 2, Supplementary Movie 1).”

Page 8, lines 2-5: “It is possible to extend the droplet transport distance by tuning the gradient steepness and length^{12,18}. Furthermore, we believe with further development, periodic gradients could be used to extend transport beyond that possible using a single gradient.”

Supplementary Fig. 2 has been revised to highlight the transport distance of a 3 μL droplet on a smooth gradient surface ($r = 1$).

- 4) *“Similarly, compared to other manipulation approaches (see above), this method seems to be a one-way form of transport. While this might be enough for some applications (water harvesting, energy generation, self-cleaning surfaces), it is a limitation for other applications of droplet transport. In fact, in several videos, it can be seen that the droplets fully wet the surface at the end of its journey due to the high wettability of the surface and, thus, do not retain their shape. The authors should clarify this point in their manuscript.”*

We have worked to demonstrate/characterize the fundamental aspects of transport on mechanically tunable chemical gradients, which at this stage only includes unidirectional transport of droplets along linear paths (as correctly noted by the referee). We agree that while this style of transport is ideal for water harvesting and self-cleaning surfaces, more sophisticated manipulations would be required for lab-on-chip type applications. We believe that the concepts demonstrated here do point out a pathway towards circumventing current limitations—for example long-range mechano-programmable droplet transport, which would incorporate periodic gradients aided by the input of additional kinetic energy (e.g., energy from mechanical vibrations, Ref. 2) to move droplets across gradient barriers—but concede that more efforts are required to achieve such high levels of control. We note that regardless of the shape, number, and organization of periodic gradients, droplet trajectory will always follow the surface energy profile, and chemical gradients will always be “one-way” tracks. We have added additional discussion to the main text in order to articulate these nuanced concepts clearly. We have made the following specific changes:

Page 7, lines 2-5: “It is possible to extend the droplet transport distance by tuning the gradient steepness and length^{12,18}. Furthermore, we believe with further development, periodic gradients could be used to extend transport beyond that possible using a single gradient.”

Beginning on page 14, line 22: “We aimed to demonstrate/characterize the fundamental aspects of dynamic droplet transport across single, one-dimensional (linear) gradients, which are ideally suited for water harvesting, anti-fouling, and self-cleaning applications. Further advancement of our system (i.e., incorporation of periodic gradients and/or energy input to aid in the transition of droplets across the surface energy barrier present at gradient boundaries) will be necessary to realize the full potential of this technology. We believe that extensions of the demonstrated concepts will lead to more sophisticated surface fluidic capabilities useful to, for example, biomedical and analytical devices, mechano-switchable water sorting devices, surface lab-on-chip devices, and adaptive materials for emergent robotic applications. For example, the use of mechano-activated, two-dimensional (non-linear), periodic gradients together with the input of additional kinetic energy (i.e., energy from mechanical vibrations)

could enable advanced lab-on-chip functionality. In this way the inherent “one-way” transport of chemical gradients could be overcome, and more intricate droplet manipulations realized.”

- 5) *“The authors modify the general expression for the force on chemical gradients first used by Daniel et al. and subsequently modified by Shastry et al. to account for surface texture. They then introduce a fitting constant to explain the discrepancy between model prediction and experimental data. Although this discrepancy has been reported by other authors, the way in which the authors correct for it in their work is purely phenomenological. I would recommend clarifying this aspect in their manuscript.”*

We have clarified that modification of the fundamental equations referenced are based exclusively on empirical observations that are system dependent. Further, in response to comment #7 below, we now explicitly define and show the fitting parameter used in modifying the referenced equations.

We have added the following to the main text:

Page 8, lines 12-13: “—one based on empirical data that is specific to the system under investigation—...”

- 6) *“In their model, the authors modify the expression for the force that has a power 2 dependence on R to account for discrepancy between prediction and experiments. They justify based on fitting the discrepancy to trends depending on R^2 , rR and R . Based on the Pearson correlation, all trends can be plausible. The authors should try to fit over a larger range of R values in order to come to a more convincing conclusion, otherwise the choice of where to introduce the fitting parameter remains arbitrary. For example, the effect could also depend on an underestimation of the hysteresis force (Eq. 2), which is often reported with a prefactor that can take values up to π depending on the exact shape of the three-phase contact line. The authors touch briefly on the topic but the argument based on the previous fitting is not convincing.”*

We acknowledge that the range of critical radii (R_c) surveyed was limited. To clarify the experimental constraints responsible for the reported range, we note the following: 1) The smallest R_c value reported corresponds to a flat surface ($r = 0$ and $\varepsilon_c = 0$) and is characteristic for the gradient profile studied and will not change unless the gradient is changed (a variable we held constant); 2) The highest R_c value reported ($R_c = 2.13$ mm) corresponded to a high compressive strain state relative to our system’s capabilities ($r = 1.19$ and $\varepsilon_c = 0.175$) which significantly deformed the surface and required large droplets to be measured. These large droplets (approaching 5 mm in diameter) were becoming commensurate with the size of the gradient itself (~ 13 mm) and increasingly susceptible to other factors (e.g., gravitational deformations) which complicated accurate observation/measurement. For these reasons we could not gather reliable data beyond $\varepsilon_c = 0.175$; and 3) The resolution of the R_c values reported

in this range ($\varepsilon_c = 0 - 0.175$) was determined by the resolution of the strain applied (which determined r). For low values of compressive strain these measurements were not trivial. That said, we do agree that there was a large gap in our experimental data between $r = 1.15$ and $r = 1.19$, so we conducted additional experiments and performed additional analyses in order to include an additional data point in this gap ($r = 1.17$ in the revised manuscript). We agreed with the referee that this additional data could help to strengthen our decision to modify the gradient term (over the contact angle hysteresis term), as we discuss further below.

In revisiting the relevant discussions on the Pearson's correlation values and the decision to modify the gradient term, we agree with the referee that clarification is required to strengthen the manuscript. To begin this discussion, we report the new correlation values calculated from the data set that included the additional data point discussed above: the correlation values changed from 0.9975 to 0.9974 for force (F) vs R_c^2 , 0.9773 to 0.9759 for F vs R_c , and 0.9725 to 0.9705 for F vs rR_c . What we can see immediately is that the correlation between F and R_c^2 remained nearly unchanged, whereas the correlation between F and R_c (and rR_c) decreased, further increasing the difference between the correlation factors. To justify the addition of the fitting parameter to the driving force term (the R^2 dependent term), we used a Z-test to determine whether the differences between the correlation values for F vs R_c^2 and F vs R_c , and F vs R_c^2 and F vs rR_c , were statistically significant. From these Z-tests we determined that at 95% confidence ($P < 0.05$) the correlation of F vs R_c^2 was statistically different to that of F vs R_c and F vs rR_c . Based on these analyses (namely that F vs R_c^2 showed the strongest correlation and that this correlation differed significantly from the others), we applied a fitting parameter to the driving force. We viewed strong agreement of the observed critical radii (Fig. 2C) with the values predicted from the modified expression as further support our decision.

The following specific changes have been made to the main text:

We updated Fig. 2b, Supplementary Fig. 6 and 7 with the new data at $r = 1.17$ and have revised the Pearson correlation values accordingly.

Page 9, line 15-21: "...and verified that the differences between the F vs. R_c^2 correlation and the other two correlations were statistically significant using a Z-test ($P < 0.05$). We viewed the significant correlation of F and R_c^2 together with the results of the Z-tests as strong support for the assertion..."

Page 9, lines 20-21: "and thus applied a fitting parameter to the positive driving force term."

Page 10, line 17: "1.77 mm"

Page 10, line 18: "0.1625"

Page 10, line 20: " 1.77 ± 0.03 mm"

- 7) *"Connected to the previous point, the authors introduce the fitting constant by renaming the contact angle gradient. I would separate the two things and just introduce a separated fitting constant as the contact angle gradient is a physically measurable parameter that depends on the material and has been measured experimentally by the authors."*

We agree with the referee's sentiments. We have removed Φ' from equation (7) and introduced a fitting constant "C" to apply the appropriate reduction to the driving force term.

We have added the following to the main text:

Page 9, line 26: "reduction in F_d ..."

Page 10, line 1: "Specifically, we introduced a fitting constant "C" into equation (4), where $C = 3.04$.

We modified equation (7) to:

$$F' = \frac{\pi}{C} R^2 \gamma \Phi - 2\gamma r R b \quad (7)$$

We have revised equation (7) in Fig. 2 accordingly.

- 8) *"Both in main text and supplementaries, the authors mention that they neglect the viscous drag force. However, this force is very fundamental for the force balance in the motion of the droplet and it's used by the same authors to obtain an expression for velocity later in the text (Eq. 5). This should be clarified. I was also unclear on how the authors go from expression 3 to expression 5. This should be clarified, and a self-standing derivation should be included with all assumption and approximations made rather than being left to the supplementary information."*

As indicated by the referee, we initially included a more detailed discussion of the viscous drag force as part of the Supplementary Information. We agree that including these critical details in the main text will strengthen the manuscript and that neglecting the only term which is velocity dependent creates confusion when the velocity relation is given. We have moved the text from the Supplementary Information to the main text and revised the main text to clarify these derivations along with any approximations that were made.

Page 4, line 24: "...prior to the onset of motion..."

Page 5, lines 7-21: "During droplet transport, a viscous drag force (F_v) acts against the microdroplet. Using the lubrication approximation and assuming a circular droplet, F_v can be approximated as:^{2,3,9,12,34}

$$F_v = 3\eta\pi RrV \int_{x_{min}}^{x_{max}} \frac{dx}{H} \approx 3\eta\pi RrV \ln \left(\frac{x_{max}}{x_{min}} \right) \quad (5)$$

where η is the viscosity of the liquid, V is the droplet velocity, and H is the height of droplet. The integration limits x_{min} and x_{max} are the characteristic lengths of a droplet (x_{min} is the length of a molecule, and x_{max} can be represented as the radius of a droplet). The steady-state velocity of a liquid microdroplet moving along a wettability gradient can then be obtained by equating F_d to F_v and approximated according to the Greenspan model as²:

$$v \cong \frac{\gamma R}{\eta} (\Phi) \quad (6)$$

Although the above treatment does not consider the influence of contact angle hysteresis on droplet velocity², experimental observations follow this approximation, and droplet velocity scales linearly with droplet radius^{2,3,7}. During incipient motion—the condition immediately prior to the onset of droplet transport critical to this work— F_v is negligible and can be ignored^{9,35}.”

"A few minor points:"

9) "On line 125, the authors mention a pre-strained PDMS film. Details on the pre-straining process should be provided."

We include detailed procedures for the fabrication of the films in the Supplementary Information (section 2). Based on this comment, we believe some of this information should be in the main text as well and have added the following to the main text:

Page 6, line 18: "...($\epsilon_{pre} = 0.20$ or 20%)..."

10) "Statistical information (number of experiments, type of error, etc) should be included in all figure captions."

We have revised the figure captions in the main text as follows:

Figure 1, page 19, lines 6-7: "(N = 30 for each data point, data reported as $\bar{x} \pm s$ where \bar{x} is the mean and s is the standard deviation)."

Figure 1, page 19, line 9: "(N = 5 for each data point, data reported as $\bar{x} \pm s$ where \bar{x} is the mean and s is the standard deviation)."

Figure 1, page 18, lines 13-14: "(N = 5 for each data point, data reported as $\bar{x} \pm s$ where \bar{x} is the mean and s is the standard deviation)."

Figure 2, page 19, lines 7-9: "(N = 5 for each data point, data reported as $\bar{x} \pm s$ where \bar{x} is the mean and s is the standard deviation). The error in force measurements were obtained following the propagation of uncertainty from values in equation (4)."

We have also revised the figure captions in the Supplementary Information document accordingly.

11) "Surfaces seem to show a curvature (probably due to the mechanical stress). Can this curvature influence the authors' results?"

We increased the thickness of PDMS films utilized in this study to minimize the influence of film curvature. As noted by the referee, the curvature is a result of mechanical tension following pre-strain release and would directly affect the wrinkled topography due to stress dissipation into the bulk material. We also held the films flat and any effects residual curvature may have on wrinkle formation and surface topography is accounted for through direct measurement of surface roughness via confocal microscopy (Figure 1c).

We have added the following to the main text:

Page 7, lines 4-5: “We minimized the effects of stress-induced substrate curvature by fabricating 1 mm thick PDMS films and keeping the films flat during characterization.”

12) *“The quantities in the supplementary shouldn't be bold unless they are vectors.”*

We have converted bolded quantities to plain text in the Supplementary Information.

Referee 2:

Note: The page and line numbers below refer to the clean version of the manuscript and S.I., not the marked versions.

From the Overview:

“The authors combined the chemical gradient and surface wrinkle to manipulate the droplet transportation on the surface. They detailed the discussion about the driving force, resistant force and the moving velocity of the droplet. While the discussion about the driving velocity could be interesting to the field, the technical design and the results are somehow not impressed. The following are detailed comments.”

We agree with the referee’s general statement that the “discussions about the driving (forces)” at play in the reported dynamic droplet transport could be of interest to the expanding community of researchers exploring technologies applicable to surface droplet manipulation. We, however, do not believe it is fair to compare the functionality possible using our approach to that available to more mature technologies (e.g., electrowetting). The current work is the first demonstration of the combination of surface chemical gradients and mechanically tunable microwrinkles toward dynamic and programmable surface droplet manipulation. More detailed discussion about the limitations this approach overcomes is available in the response to Referee 1 (see comment #1). We intended to focus on the physical underpinnings and the most basic functions in this report as it paves the way to more sophisticated applications (more details may be found in response to Referee 1, comment #6).

1) *“The motivation of the current study is somehow unclear, the combination of several techniques cannot be reasoned as the motivation. I noticed that the references are rather old, and many important new works are missing. For instance, Nature Materials, 2019,18, 236, which can transport a droplet at a very large speed and to a long distance, and even on a upside down surface. Sci. Adv. 2017, 3, eaao3530 also reported a surface design for guiding the water transportation; Adv. Sci. 2020, 2001650 reported a surface presents a similar strategy using stress to regulate the surface pattern and then the state-switching of droplet on the surface. Actually, there are several more works making use of the stress.”*

We note that similar concerns were raised by Referee 1 (see comment #1 and #2). We significantly modified the introduction to articulate the motivations and advantages of our approach clearly. For full details concerning the modifications we made, we direct Referee 2 to our reply on comments #1 and #2 from Referee 1. In addition to those modifications, we

have also cited the suggested articles published in *Nature Mater.* by Sun et al. (Ref. 8) and *Sci. Adv.* by Li et al. (Ref. 17). The specific references follow:

8. Sun, Q. *et al.* Surface charge printing for programmed droplet transport. *Nat. Mater.* **18**, 936–941 (2019).
17. Li, J. *et al.* Topological liquid diode. *Sci. Adv.* **3**, eaao3530 (2017).

- 2) *“How was the wrinkle prepared, if an unidirectional stress was used, then the wrinkles have a certain orientation, it will have a strong influence of the shape of the droplet on it, then, how was the radius of the droplet determined? From the video, we can see that many droplets are extremely asymmetric. Due to the fabrication process, any anisotropic properties of the droplets? Like the contact angle, and also the transportation behaviors?”*

We prepared the wrinkles following the general procedure outlined in Fig. 1a. This procedure resulted in a wrinkled surface with a sinusoidal profile that was along the axis of strain (Supplementary Fig. 1). In this work, we always oriented the chemical gradients along the axis of strain (Fig. 1a) and measured droplet radii and droplet transport along this direction. We have clarified this point in the text. We agree that anisotropy is a possibility on these surfaces but note that it is most significant under high wettability conditions or when the droplets are moving. For this reason, we conducted all measurements to probe the conditions for the onset of droplet motion at the hydrophobic end of the gradient when droplets are not moving (due to roughness) and anisotropy is minimal. We have clarified this point as well. We agree that investigating gradient orientations that are not necessarily aligned with the wrinkle profile represents an interesting control vector but chose to probe the most straightforward construct in this initial demonstration. The following specific changes have been made:

Page 8, line 19: “... and measured R_c along the axis of strain.”

Page 9, lines 8-10: “In order to minimize the effects of droplet anisotropy due to the wrinkled topography, R_c was measured exclusively at the hydrophobic end of the chemical gradient where anisotropy was nominal.”

- 3) *“P5, Quantitative characterization about the surfaces, like the structure parameters, surface energy and the gradients should be given in the main text, which are very important for the following discussions.”*

We have added two additional panels to Figure 1. Specifically, we now include characterization of the surface topography (previously Supplementary Fig. 1) and surface wettability of the chemical gradient (previously Supplementary Fig. 2).

We have modified the main text and Supplementary Information documents accordingly to account for these changes.

- 4) *“How is the state of the droplet on the surface, it is in Cassie or Wenzel state? Both case fit to the model proposed?”*

Given the low contact angle of the droplets and the measured characteristics of the surface topography (wrinkle amplitude, roughness, etc.), we expected that the droplets would take on

the Wenzel state (Ref. 22, 33, 34). We observed conformal interfacial contact of droplets in our studies (e.g., see Supporting Fig. 1 and 3), which agreed with this expectation of the Wenzel wetting state. It is true that the model we adopted could be modified to account for a Cassie Baxter wetting state, but equations (4) and (7) would need to include the contact fraction of the droplet to the solid and air interfaces. We did not include these modifications as our studies followed the Wenzel state.

We have revised the manuscript as follows:

Page 5 lines 2-5: “The wetting state of a droplet on a wrinkled surface (where $r > 1$) depends on the wrinkle characteristics (amplitude, wavelength, etc.)²² and for droplets in the Wenzel wetting state (observed in this work), contact angle hysteresis scales with surface roughness^{33,34}.”

- 5) *“Fig2a, the authors discussed the critical size of the droplets, however, without the precise description of the surface, it is meaningless. When a larger droplet is placed on the surface, surely, it will have a larger driving force as a larger contact area and thus a larger gradient exists.”*

The critical radius (R_c) of the droplets at different states of compressive strain was measured and reported in Figure 2b (we assume this panel is what is being referenced). We agree with the referee that details on the surface are critical to understanding and predicting R_c in our system. We included all the relevant details in our original manuscript, though they were not as centralized as possible. We thus, in response to this comment and comment #3, have updated Fig. 1 to include relevant surface characterizations. Furthermore, we also included (in the original and revised manuscript) measurements of the surface gradient under compressive strain (Supplementary Fig. 1) but have elected to leave these details in the Supplementary Information. If the referee feels these details should also be included in Fig. 1, we are happy to make this modification.

As for the relationship between these surface parameters and radius we included (in the original and revised manuscripts) the governing equations. We note that they have been slightly modified in the revision to respond to comments from Referee 1. Specifically, the gradient profile (Φ), which we held constant in this study, and its relationship to force and radius (R) is accounted for by equations (4) and (7) of the main text (also included below).

$$F = \pi R^2 \gamma \Phi - 2\gamma r R b \quad (4)$$

$$F' = \frac{\pi}{c} R^2 \gamma \Phi - 2\gamma r R b \quad (7)$$

These equations encapsulate the impact of droplet radius (R) and surface roughness (r) on droplet transport and our observations were fully consistent with them (e.g., as roughness increased the critical radius for transport also increased in a predictable fashion, Fig. 2c).

- 6) *“The most fundamental capability stemming from this concept is switching transport “on” from an “off” state”, however, this kind of switching has been widely reported. like Adv. Sci.*

2020, 2001650; Adv. Mater. 2017, 29; Adv. Funct. Mater. 2018, 28, 1800625 et al. Is the switching between two states reversible?

We directly demonstrated reversibility in Fig. 3e, Fig. 4b, and Fig. 4e. We agree with the referee that reversibility is a key advancement critical to the significance of this report. We focused on demonstrating switchability by toggling transport “on” from an “off” state and back to an “on” state. Further, we highlighted the ability to toggle transport “off” from and “on” state in Fig. 3e.

We have revised the main text to further stress the importance of reversible droplet transport:

Page 11, beginning on line 10 “Extension of this new capability enables programmable and reversible switching of the surface... (Fig. 3f).”

Page 13, lines 2-3: “...reversible programming of microdroplet transport...”.

REVIEWERS' COMMENTS

Reviewer #1 (Remarks to the Author):

I thank the authors for the revised version of the manuscript. All my comments have been taken into account and I am satisfied with the new version of the article. This work presents a new approach to the controllable transport of droplets on surfaces and, although there is still room for development, I believe that the results are novel and original enough to deserve publication in Nature Communications.

Reviewer #2 (Remarks to the Author):

The authors answered part of my concerns. From the title and the results, we know that the mechanical regulation of the surface is very important for this work. There are at least a dozen of works, part of which have been mentioned in my last comments, reporting on the regulation of surface wettabilities by mechanical regulation of the surfaces. Though these works are not exactly the same as the work here, these previous works should be stated in the introduction part as the background. This is a very important for the audience to understand the novelty and the advantage of this work. However, the authors refuse to modify the introduction part, therefore, I would suggest to reject the work.

Referee 1:

Note: The page and line numbers below refer to the clean version of the manuscript and S.I., not the marked versions.

“I thank the authors for the revised version of the manuscript. All my comments have been taken into account and I am satisfied with the new version of the article. This work presents a new approach to the controllable transport of droplets on surfaces and, although there is still room for development, I believe that the results are novel and original enough to deserve publication in Nature Communications.”

We appreciate the time and effort of Referee 1 whose thoughtful comments aided us in significantly strengthening the manuscript. We agree that there is room for development and look forward to reporting our future efforts which stem from this critical initial report on mechano-adaptive droplet transport.

Referee 2:

Note: The page and line numbers below refer to the clean version of the manuscript and S.I., not the marked versions.

“The authors answered part of my concerns. From the title and the results, we know that the mechanical regulation of the surface is very important for this work. There are at least a dozen of works, part of which have been mentioned in my last comments, reporting on the regulation of surface wettabilities by mechanical regulation of the surfaces. Though these works are not exactly the same as the work here, these previous works should be stated in the introduction part as the background. This is a very important for the audience to understand the novelty and the advantage of this work. However, the authors refuse to modify the introduction part, therefore, I would suggest to reject the work.”

We strongly disagree with the Referee’s characterization of our first revision. We worked to provide a thorough and complete revision which addressed each comment and concern raised by Referee 2. We included two additional references on topics addressing the mechano-regulation of surface wettability suggested by the Referee and believed that these specific changes adequately addressed the Referee’s general concerns. We did not refuse to modify the introduction, rather, we extensively revised/rewrote this section (and others) in our revised manuscript in an effort to address the concerns of Referee 2. It would seem that, given the Referee’s strong stance against our work in the above comments, we were not deliberate enough in acknowledging the prior works on mechanical regulation of surface wettability (which we view as related but different than transport) in our revision. We therefore have included another two references on the topic and explicitly reference them as “mechanically reversible” strategies. We hope this further emphasis satisfies the concerns of Referee 2.

The following specific changes were made:

Page 3, lines 21-22: “Mechanically reversible surface topographies have been used to modulate surface wettability^{25,28,29} and adhesion^{30,31} ...”

We have added reference # 29 and 31.

29. Wang, J.-N. et al. Wearable superhydrophobic elastomer skin with switchable wettability. *Adv. Funct. Mater.* **28**, 1800625, (2018).
31. Li. Q. et al. Reversible structure engineering of bioinspired anisotropic surface for droplet recognition and transportation. *Adv. Sci.* **7**, 2001650, (2020).